# Graph Regression Model for Spatial and Temporal Environmental Data—Case of Carbon Dioxide Emissions in the United States

**DOI:** 10.3390/e25091272

**Published:** 2023-08-29

**Authors:** Roméo Tayewo, François Septier, Ido Nevat, Gareth W. Peters

**Affiliations:** 1Univ Bretagne Sud, CNRS UMR 6205, LMBA, F-56000 Vannes, France; francois.septier@univ-ubs.fr; 2TUMCREATE, 1 Create Way, #10-02 CREATE Tower, Singapore 138602, Singapore; ido.nevat@tum-create.edu.sg; 3Department of Statistics and Applied Probability, University of California Santa Barbara, Santa Barbara, CA 93106, USA; garethpeters@ucsb.edu

**Keywords:** graph regression model, spatio-temporal data, CO_2_ emission

## Abstract

We develop a new model for spatio-temporal data. More specifically, a graph penalty function is incorporated in the cost function in order to estimate the unknown parameters of a spatio-temporal mixed-effect model based on a generalized linear model. This model allows for more flexible and general regression relationships than classical linear ones through the use of generalized linear models (GLMs) and also captures the inherent structural dependencies or relationships of the data through this regularization based on the graph Laplacian. We use a publicly available dataset from the National Centers for Environmental Information (NCEI) in the United States of America and perform statistical inferences of future CO${}_{2}$ emissions in 59 counties. We empirically show how the proposed method outperforms widely used methods, such as the ordinary least squares (OLS) and ridge regression for this challenging problem.

## 1. Introduction

Statistical models for spatio-temporal data are invaluable tools in environmental applications, providing insights, predictions, and actionable information for understanding and managing complex environmental phenomena [1]. Such models help uncover complex patterns and trends, providing insights into how environmental variables change geographically and temporally. Many environmental datasets are collected at specific locations and times, leaving gaps in information. Statistical models help interpolate and map values between observation points, providing a complete spatial and temporal picture of the phenomenon being studied. Moreover, environmental applications frequently require predicting future values or conditions. Statistical models allow for accurate predictions by capturing the spatial and temporal dependencies present in the data. Such predictions provided by these models provide valuable information for decision makers by quantifying the effects of various factors on the environment and projecting the consequences of different actions.

Let $\left\{y_{t,s}:s\in \Omega_{s},t\in \Omega_{t}\right\}$ denote the spatio-temporal random process for a phenomenon of interest evolving through space and time. As an example, $y_{t,s}$ might be the CO${}_{2}$ emission level at a geographical coordinate s=(latitude,longitude) on the sphere at a given time *t*. Traditionally, one considers models for such a process from a *descriptive* context, primarily in terms of the first few moments of a probability distribution (i.e., mean and covariance functions in the case of a Gaussian process). Descriptive models are generally based on the *spatio-temporal mixed-effect model* [1,2], in which the spatio-temporal process is described with a deterministic mean function and some random effects capturing the spatio-temporal variability and interaction:(1)$$y_{t,s}=\mu_{t,s}+\epsilon_{t,s}$$
where $\mu_{t,s}$ is a deterministic (spatio-temporal) mean function or trend, and $\epsilon_{t,s}$ a zero-mean random effect, which generally depends on some finite number of unknown parameters. A common choice for the trend is to consider the following linear form $\mu_{t,s}=\phi_{t,s}\beta$, where $\phi_{t,s}$ represents a vector of known covariates and β a set of unknown coefficients. Generally, with such a model, it is generally assumed that the process of interest is Gaussian. However, in real-world scenarios, data can exhibit heavy tails or outliers, which can significantly affect the distribution’s shape and parameters. If these extreme values are not accounted for, it can lead to biased estimates and incorrect inferences. As a consequence, a more advanced model based on a generalized linear model (GLM) has been proposed [3]. The systematic component of the GLM specifies a relationship between the mean response and the covariates through a possibly nonlinear but known link function. Note that some additional random effects can be added in the transformed mean function, leading to the so-called *generalized linear mixed model* (GLMM) [4].

The main challenge of such models lies in estimating the unknown parameters. Once this important step is done, the different tasks of interest (prediction, decision, etc.) can be performed. Unfortunately, the inference of these parameters can lead to overfitting, multicollinearity-related instability, and lack of variable selection, resulting in complex models with high variance. As a consequence, regularization methods using the $\ell_{1}$ and/or $\ell_{2}$ norm as penalty function are generally used in practice to mitigate these issues by controlling the model complexity, improving generalization, and enhancing the stability of coefficient estimates [5,6].

### Contributions

Graph signal processing is a rapidly developing field that lies at the intersection between signal processing, machine learning and graph theory. In recent years, graph-based approaches to machine learning problems have proven effective at exploiting the intrinsic structure of complex datasets [7]. Recently, graph penalties were applied successfully to the reconstruction of a time-varying graph signal [8,9] or to the regression with a simple linear model [10,11]. In these works, the results highlight that regularization based on the graph structure could have an advantage over more traditional norm-based ones in situations where the data or variables have inherent structural dependencies or relationships. The main advantage of graph penalties is that they take into account the underlying graph structure of the variables, capturing dependencies and correlations that might not be adequately addressed by norm-based penalties.

In this work, we propose a novel and general spatio-temporal model that incorporates a graph penalty function in order to estimate the unknown parameters of a spatio-temporal mixed-effect model based on a generalized linear model. In addition, different structures of graph dependencies are discussed. Finally, the proposed model is applied to a real and important environmental problem: the prediction of CO${}_{2}$ emissions in the United States. As recently discussed in [12], regression analysis is one of the most widely used statistical method to characterize the influence of selected independent variables on a dependent variable and thus has been widely used to forecast CO${}_{2}$ emissions. To the best of our knowledge, this is the first time that a more advanced model, i.e., a GLM-based spatio-temporal mixed effect model with graph penalties, is proposed to predict CO${}_{2}$  emissions.

## 2. Problem Statement—The Classical Approach

In this section, we first provide a background of graphs and their properties, then we introduce the system model of our problem followed by the classical approach which uses a linear regression structure.

### 2.1. Preliminaries

Let us consider a weighted, undirected graph G=(V,E,A) composed of |V|=N vertices. $A\in R^{N\times N}$ is the weighted adjacency matrix, where $A_{ij}\geq 0$ represents the strength of the interaction between nodes *i* and *j*. An example of such a graph is depicted in Figure 1. E is the set of edges, and therefore (i,j)∈E implies $A_{ij}>0$ and (i,j)∉E implies $A_{ij}=0$. The graph can be defined through the (unnormalized) Laplacian matrix $L\in R^{N\times N}$:(2)L=D−A
where D corresponds to the degree matrix of the graph as $D=\mathrm{diag}(D_{11},D_{22},\ldots D_{NN})$, where $D_{ii}$ is the *i*-th column sum (or row sum) of the adjacency matrix *A*.

The graph Laplacian, closely related to the continuous domain Laplace operator, has many interesting properties. One of them is the ability to inform about the connectedness of the graph. By combining this property with any graph signal at time *t*, $y_{t}\in R^{N}$, in the following quadratic sum,
(3)$$y_{t}^{⊤}Ly_{T}=\sum_{\left(i,j\right)}A_{ij}{\left(y_{t,i}-y_{t,j}\right)}^{2}$$
can be considered a measure of the cross-sectional similarity of the signal, with smaller values indicating a smoother signal reaching a minimum of zero for a function that is constant on all connected sub-components [13].

### 2.2. System Model

The main objective of this paper is to design a statistical regression model in order to characterize and predict CO${}_{2}$ emissions across time and space. More precisely, the paper is concerned with the situation where a signal $y_{t}={\left(y_{t,1},y_{t,2},\ldots ,y_{t,N}\right)}^{⊤}\in R^{N}$ is measured on the vertices of a fixed graph at a set of discrete times t∈[1,2,…,T]. This vector corresponds to the CO${}_{2}$ emission measured at *N* different spatial locations at time *t*. At each of these time instants, a vector of *K* covariates $x_{t}\in R^{K}$ is also measured, which is not necessarily linked to any node or set of nodes.


Objectives:
1.Determine, for each of the *N* different locations, the specific relationship between the response variables ${\left\{y_{t,i}\right\}}_{t=1}^{T}$ and the set of covariates ${\left\{x_{t}\right\}}_{t=1}^{T}$.2.Based on this relationship, make a prediction of the CO${}_{2}$ levels in different locations in space and time.


### 2.3. Problem Formulation with a Classical Linear Regression Model

The most common form of structural assumption is that the responses are assumed to be related to predictors through some deterministic function *f* and some additive random error component $\epsilon_{i}$ so that for the *i*-th location and ∀t=1,…,T we have that
(4)$$y_{t,i}=f_{i}\left(x_{t}\right)+\epsilon_{i},$$
where $\epsilon_{i}$ is a zero-mean error random variable. Therefore, a classical procedure consists of approximating the true function $f_{i}$ by a linear combination of basis functions:(5)$$f_{i}\left(x_{t}\right)\approx \sum_{p=1}^{P}\beta_{i,p}\phi_{i,p}\left(x_{t}\right)=\phi_{i}{\left(x_{t}\right)}^{T}\beta_{i},$$
where $\beta_{i}={\left[\begin{array}{c} \beta_{i,1}\ldots \beta_{i,P} \end{array}\right]}^{T}$ is the set of coefficients corresponding to basis functions $\phi_{i}\left(x_{t}\right)={\left[\begin{array}{c} \phi_{i,1}\left(x_{t}\right)\ldots \phi_{i,P}\left(x_{t}\right) \end{array}\right]}^{T}$ in order to approximate the function $f_{i}\left(\;\cdot \;\right)$ associated to the signal over time at *i*-th location, i.e., ${\left\{y_{t,i}\right\}}_{t=1}^{T}$.

The linear regression model over all the *N* different locations could be formulated in a matrix form as follows ∀t∈[1,2,…,T]:(6)$$y_{t}=\Phi_{t}\beta +\epsilon_{t},$$
where
(7)$$\Phi_{t}=\left[\begin{array}{cccc} \phi_{1}\left(x_{t}\right) & 0_{1\times P} & \ldots & 0_{1\times P}\\ 0_{1\times P} & \phi_{2}\left(x_{t}\right) & & \vdots \\ \vdots & \; & \ddots & 0_{1\times P}\\ 0_{1\times P} & \ldots & 0_{1\times P} & \phi_{N}\left(x_{t}\right) \end{array}\right]\;\beta =\left[\begin{array}{c} \beta_{1}\\ \vdots \\ \beta_{N} \end{array}\right]\;\mathrm{and}\;\epsilon_{t}=\left[\begin{array}{c} \epsilon_{t,1}\\ \vdots \\ \epsilon_{t,N} \end{array}\right]$$

As a consequence, this linear regression can be fully summarized as
(8)y=Φβ+ϵ,
where $y={\left(y_{1}^{T}y_{2}^{T}\ldots \;y_{T}^{T}\right)}^{⊤}\in R^{NT\times 1}$ and
$$\Phi =\left[\begin{array}{c} \Phi_{1}\\ \vdots \\ \Phi_{T} \end{array}\right]\in R^{NT\times NP}\;\;\mathrm{and}\;\;\epsilon =\left[\begin{array}{c} \epsilon_{1}\\ \vdots \\ \epsilon_{T} \end{array}\right]\in R^{NT\times 1},$$
where $E\left[\epsilon \right]=0_{NT\times 1}$ and $\mathrm{Var}\left(\epsilon \right)=\Sigma$.

In such a model, the most common approach to estimate the regression coefficients is the generalized least square (GLS) method, which aims at minimizing the squared Mahalanobis distance of the residual vector:(9)$${\hat{\beta }}_{\mathrm{GLS}}=\underset{\beta }{arg min}{\left(y-\Phi \beta \right)}^{T}\Sigma^{-1}\left(y-\Phi \beta \right).$$

**Theorem 1** (Aitken [14]). *Consider that the following conditions are satisfied:*
*(A1)* 
*The matrix *
**Φ**
* is nonrandom and has full rank, i.e., its columns are linearly independent,*
*(A2)* 
*The vector y is a random vector such that the following hold:*
*(i)* 
*$E\left[y\right]=\Phi \beta_{0}$ for some $\beta_{0}$;*
*(ii)* 
*$\mathrm{Var}\left(y\right)=\Sigma$ is a known positive definite matrix.*

*Then, the generalized least square estimator from *(Equation 9)* is given by*$${\hat{\beta }}_{\mathrm{GLS}}={\left(\Phi^{T}\Sigma^{-1}\Phi \right)}^{-1}\Phi^{T}\Sigma^{-1}y.$$
*Moreover, ${\hat{\beta }}_{\mathrm{GLS}}$ corresponds to the best linear unbiased estimator for $\beta_{0}$ and its covariance matrix is $\mathrm{Var}\left({\hat{\beta }}_{\mathrm{GLS}}\right)={\left(\Phi^{T}\Sigma^{-1}\Phi \right)}^{-1}$.*


Let us remark that the ordinary least square (OLS) estimator is nothing but a special case of the GLS estimator. They are indeed equivalent for any diagonal covariance matrix $\Sigma =\sigma^{2}I$.

### 2.4. Generalized Linear Models

In this paper, we propose to use the generalized linear model (GLM) structure [15], which is a flexible generalization of linear regression model discussed previously. In this model, the additivity assumption of the random component is removed and more importantly, the response variables can be distributed from more general distributions in the standard linear model for which one generally assumes normally distributed responses, see discussions in [16,17]. The likelihood distribution of the response variables $f_{Y}\left(y|\beta \right)$ is a member of the *exponential family*, which includes the normal, binomial, Poisson and gamma distributions, among others.

Moreover, in a GLM, a smooth and invertible function g(·), called *link function*, is introduced in order to transform the expectation of the response variable, $\mu_{t,i}\equiv E\left[y_{t,i}\right]$
(10)$$g\left(\mu_{t,i}\right)=\eta_{t,i}=\phi_{i}{\left(x_{t}\right)}^{T}\beta_{i}.$$

Because the link function is invertible, we can also write
(11)$$\mu_{t,i}=g^{-1}\left(\eta_{t,i}\right)=g^{-1}\left(\phi_{i}{\left(x_{t}\right)}^{T}\beta_{i}\right),$$
and, thus, the GLM may be thought of as a linear model for a transformation of the expected response or as a nonlinear regression model for the response. In theory, the link function can be any monotonic and invertible function. The inverse link $g^{-1}$ is also called the *mean function*. Commonly employed link functions and their inverses can be found in [15]. Note that the *identity link* simply returns its argument unaltered $\mu_{t,i}=g^{-1}\left(\eta_{t,i}\right)=\eta_{t,i}=\phi_{i}{\left(x_{t}\right)}^{T}\beta_{i}$ and therefore is equivalent to the assumption (A2)-(i) of Theorem 1 used in the classical linear model.

In GLM, due to the nonlinearity induced by the link function, the regression coefficients are generally obtained with the maximum likelihood technique, which is equivalent to minimizing a cost function defined as the negative log-likelihood function $f_{Y}\left(y|\beta \right)$ as [16]
(12)$$\hat{\beta }=\underset{\beta }{arg min}V\left(y;\beta \right),$$
with $V\left(y;\beta \right)=-\ln f_{Y}\left(y|\beta \right)$.

## 3. Proposed Graph Regression Model

In this section, we develop our *penalized regression model over graph*. We first show how to overcome some of the deficiencies in traditional regression models by introducing new penalty terms which regulate the solution. Finally we provide details regarding the estimation procedure and the algorithm we develop.

### 3.1. Penalized Regression Model over Graph

In the previous section, we introduced a flexible generalization in order to model our spatial and temporal response variables of interest. Unfortunately, two main issues could arise. On the one hand, the solution of the optimization problem defined in (Equation 12) may not be unique if Φ has full rank deficiency or when the number of regression coefficients exceeds the number of observations (i.e., NP>NT). On the other hand, the learned model could suffer from poor generalization due to, for example, the choice of an overcomplicated model. To avoid such problems, the most commonly used approach is to introduce a penalty function in the optimization problem to further constrain the resulting solution as
(13)$$\hat{\beta }=\underset{\beta }{arg min}\left(V\left(y;\beta \right)+h\left(\beta ;\gamma \right)\right).$$

The penalty term h(β;γ) can be decomposed as the sum of *p* penalty functions and therefore depends on some positive tuning parameters ${\left\{\gamma_{i}\right\}}_{i=1}^{p}$ (regularization coefficients), which controls the importance of each elementary penalty function in the resulting solution. When every parameter is null, i.e., ${\left\{\gamma_{i}\right\}}_{i=1}^{p}=0$, we obtain the classical GLM solution in (Equation 12). On the contrary, for large values of γ, the influence of the penalty term on the coefficient estimate increases. The most commonly used penalty functions are the $\ell_{2}$ norm (ridge), $\ell_{1}$ norm (LASSO) or a combination of both (Elastic-net)—see [18] for details.

In this paper, we propose to use an elementary penalty function, which takes into account the specific graph structure of the observations. As in [10,11], a penalty function can be introduced in order to enforce some smoothness of the predicted mean of the signal $E\left[y_{t}\right]$ over the underlying graph at each time instant. More specifically, we propose to use the following estimator:(14)$$\begin{array}{c} \begin{array}{cc} \hat{\beta } & =\underset{\beta }{arg min}\left(V\left(y;\beta \right)+\gamma_{1}\beta^{⊤}\beta +\gamma_{2}\sum_{t=1}^{T}E{\left[y_{t}\right]}^{⊤}LE\left[y_{t}\right]\right)\\ \; & =\underset{\beta }{arg min}\left(V\left(y;\beta \right)+\gamma_{1}\beta^{⊤}\beta +\gamma_{2}\sum_{t=1}^{T}g^{-1}{\left(\phi (x_{t})\beta \right)}^{⊤}Lg^{-1}\left(\phi (x_{t})\right)\beta \right)\\ \; & =\underset{\beta }{arg min}\left(V\left(y;\beta \right)+\gamma_{1}\beta^{⊤}\beta +\gamma_{2}g^{-1}\left(\beta^{⊤}\Phi^{⊤}\right)\left(I_{T}⊗L\right)g^{-1}\left(\Phi \beta \right)\right), \end{array} \end{array}$$
where the function $g^{-1}\left(\;\cdot \;\right):R^{NT}\mapsto R^{NT}$ corresponds to the element-wise application of the inverse link function introduced in (Equation 11) on the input argument. $I_{T}⊗L$ stands for the tensor product between the identity matrix of size *T* ($I_{T}$) and the Laplacian matrix of the underlying graph (L). The penalty function is therefore the sum of two elementary ones with $\gamma_{1},\gamma_{2}\geq 0$, their regularization coefficients. The regularization $\beta^{⊤}\beta ={∥\beta ∥}^{2}$ imposes some smoothness conditions on possible solutions, which also remain bounded. Finally, the regularization based on the graph Laplacian *L* enforces the expectation of the response variable through the GLM model to be smooth over the considered graph G at each time *t*. It comes from the property of the Laplacian matrix discussed in Section 2.1.

As recently discussed in both [8,9], in some practical applications, the reconstruction of a time-varying graph signal can be significantly improved by adequately exploiting the correlations of the signal in both space and time. The authors show from several real-world datasets that the time difference signal (i.e., $E\left[y_{t}\right]-E\left[y_{t-1}\right]$ in our case) exhibits smoothness on the graph, even if signals $E\left[y_{t}\right]$ are not smooth on the graph. The proposed model can be simply rewritten as follows in order to take into account this property:(15)$$\begin{array}{c} \begin{array}{cc} \hat{\beta } & =\underset{\beta }{arg min}\left(V\left(y;\beta \right)+\gamma_{1}\beta^{⊤}\beta +\gamma_{2}g^{-1}\left(\beta^{⊤}\Phi^{⊤}\right)\tilde{L}g^{-1}\left(\Phi \beta \right)\right), \end{array} \end{array}$$

With this general formulation, several cases can be considered:*Case 1—*$\tilde{L}=I_{T}⊗L$: the penalization induces the smoothness of the successive mean vectors $E\left[y_{1}\right],\ldots ,E\left[y_{T}\right]$ over a static graph structure L.*Case 2—*$\tilde{L}=\mathrm{diag}\left(L_{1},\ldots ,L_{T}\right)$: the penalization induces the smoothness of the successive mean vectors $E\left[y_{1}\right],\ldots ,E\left[y_{T}\right]$ over a time-varying graph structure, $L_{1},\ldots ,L_{T}$.*Case 3—*$\tilde{L}=D_{h}^{⊤}\left(I_{T-1}⊗L\right)D_{h}$ or $\tilde{L}=D_{h}^{⊤}\mathrm{diag}\left(L_{1},\ldots ,L_{T-1}\right)D_{h}$: The penalization induces the smoothness of the time difference mean vectors $E\left[y_{2}\right]-E\left[y_{1}\right],\ldots ,E\left[y_{T}\right]-E\left[y_{T-1}\right]$ over a graph structure which could be either static or time varying, respectively. The matrix $D_{h}^{⊤}$ of dimension NT×N(T−1) defined as
$$D_{h}^{⊤}=\left[\begin{array}{cccccc} -I_{N} & 0_{N} & \ldots & \ldots & \ldots & 0_{N}\\ I_{N} & -I_{N} & 0_{N} & \ldots & \ldots & 0_{N}\\ 0_{N} & I_{N} & -I_{N} & 0_{N} & \ldots & 0_{N}\\ \vdots & \ddots & \ddots & \ddots & \ddots & \vdots \\ 0_{N} & \ldots & 0_{N} & I_{N} & -I_{N} & 0_{N}\\ 0_{N} & \ldots & \ldots & 0_{N} & I_{N} & -I_{N}\\ 0_{N} & \ldots & \ldots & \ldots & 0_{N} & I_{N} \end{array}\right],$$
allows to transform the mean vector into the time difference mean vector.

**Proposition 1.** 
*When the response variables are considered to be normally distributed, i.e., $y∼N\left(\Phi \beta ,\Sigma \right)$, then the solution that minimizes the cost function defined in Equation (Equation 15) is given by*

(16)
$$\hat{\beta }={\left(\Phi^{⊤}\Sigma^{-1}\Phi +\gamma_{1}I_{NP}+\gamma_{2}\Phi^{⊤}\tilde{L}\Phi \right)}^{-1}\Phi^{⊤}\Sigma^{-1}y$$



**Proof.** See Appendix A.    □

### 3.2. Learning and Prediction Procedure

As discussed in the previous section, our proposed estimator in (Equation 15) results from a regression model with a penalization function over the graph, which depends on some hyperparameters, i.e., $\gamma =\left\{\gamma_{1},\gamma_{2}\right\}$. Cross-validation techniques are the most commonly used strategies for the calibration of such hyperparameters, as they allow us to obtain an estimator of the generalization error of a model [19]. In this paper, a cross-validation technique is used by partitioning the dataset into train, validation and test sets. Only the train and validation sets are used to obtain the selected parameters/hyperparameters set. Finally, the model with the selected set is evaluated using the test set.

Cross validation (CV) is a resampling method that uses different portions of the data to test and train a model through different iterations. Resampling may be useful while working with iid data. However, as opposed to the latter, time-series data usually posses temporal dependence, and therefore, one should respect the temporal structure while performing CV in that context. To that end, we follow the procedure of forward validation (we refer to it as time series CV) originally due to [20]. More specifically, the dataset is partitioned as follows $D_{train}={\left\{x_{t},y_{t}\right\}}_{t=1}^{\rho_{train}T}$, $D_{val}={\left\{x_{t},y_{t}\right\}}_{t=\rho_{train}T+1}^{(\rho_{train}+\rho_{val})T}$ and $D_{test}={\left\{x_{t},y_{t}\right\}}_{t=(\rho_{train}+\rho_{val})T+1}^{T}$, where $\rho_{train}$ and $\rho_{val}$ correspond to the percentage of the dataset used for training and validation, respectively. In this paper, we set $\rho_{val}=\frac{1-\rho_{train}}{2}$ to have the same number of data in both the validation and test sets. The set of hyperparameters and parameters are obtained by minimizing the generalization error approximated using the validation set. In practice, the hyperparameters are optimized using either numerical optimization methods that do not require a gradient (e.g., Nelder–Mead optimizer) or a grid of discrete values. The proposed learning procedure used in this work is summarized in Algorithm 1.
**Algorithm 1** Learning procedure of the proposed penalized regression model over graph**Input:** $D_{train}={\left\{x_{t},y_{t}\right\}}_{t=1}^{\rho_{train}T}$,            $D_{val}={\left\{x_{t},y_{t}\right\}}_{t=\rho_{train}T+1}^{(\rho_{train}+\rho_{val})T}$            $D_{test}={\left\{x_{t},y_{t}\right\}}_{t=(\rho_{train}+\rho_{val})T+1}^{T}$1:Iterations of a numerical optimization method2:**while** 
$E_{D_{val}}^{*}\neq E_{D_{val}}^{min}$
 **do**3:    Let $\gamma^{*}$ denote the candidate for the values of hyperparameters for this iteration of the chosen derivative-free optimization technique.4:    Given $\gamma^{*}$, obtain the optimal regression coefficient ${\hat{\beta }}^{*}$ in (Equation 15) using only the data from the training set $D_{train}$:
$${\hat{\beta }}^{*}=\underset{\beta }{arg min}\left(V\left(y\in D_{train};\beta \right)+\gamma_{1}^{*}\beta^{⊤}\beta +\gamma_{2}^{*}\sum_{t\in D_{train}}g^{-1}{\left(\phi (x_{t})\beta \right)}^{⊤}\tilde{L}g^{-1}\left(\phi (x_{t})\right)\beta \right).$$
either by a numerical optimization technique or Equation (Equation 16) in case of Gaussian likelihood.5:    Compute the estimator of the generalization error using the validation set:
$$E_{D_{val}}^{*}=\frac{1}{\rho_{val}T}\sum_{t\in D_{val}}\left|\right|y_{t}-g^{-1}\left(\phi (x_{t})\right){\hat{\beta }}^{*}{\left|\right|}^{2}$$6:**end while****Output**: Optimal hyperparameters $\hat{\gamma }$ and regression coefficients $\hat{\beta }$

## 4. Numerical Study—CO${}_{2}$ Prediction in the United States

In this section, we empirically assess the benefit of using our proposed penalized regression model over graph for the prediction of CO${}_{2}$ in the United States. For this purpose, the CO${}_{2}$ emission levels were obtained from the Vulcan project (https://vulcan.rc.nau.edu/ (accessed on 1 August 2023)) [21] and more especially the dataset (https://daac.ornl.gov/cgi-bin/dsviewer.pl?ds_id=1810 (accessed on 1 August 2023)), which provides emissions on a 1 km by 1 km regular grid with an hourly time resolution for the 2010–2015 time period. More specifically, the response variable vector $y_{t}$ corresponds to the CO${}_{2}$ emissions for the *t*-th day after 1 January 2011 at N=59 different counties on the east coast of the United States of America (see Appendix B for the full list of selected counties).

On the other hand, among the explanatory variables presented in detail below, there are weather data from weather daily information available on the platform https://www.ncdc.noaa.gov/ghcnd-data-access (accessed on 1 August 2023) of National Centers for Environmental Information (NCEI) in the United States of America. NCEI manages one of the largest archives of atmospheric, coastal, geophysical, and oceanic research in the world.

### 4.1. Choice of Covariates and Data Pre-Processing

The covariates we propose to use to model the daily CO${}_{2}$ emissions at the US counties level are composed of three types of data:Daily weather data (available on the platform of National Centers for Environmental Information (NCEI) https://www.ncdc.noaa.gov/ghcnd-data-access (accessed on 1 August 2023)) in the United States of America including maximal temperature (*TMAX*), minimal temperature (*TMIN*) and precipitation (*PREC*);Temporal information to capture the time patterns of the data;Lagged CO${}_{2}$ emission variables to take into account the time correlation of the response.

All the variables related to the first two points are commonly used as covariates for each county, whereas lagged variables are county-specific.

Firstly, for the weather data, a number of steps are taken to pre-process them before feeding into the learning procedure described in Algorithm 1. Firstly any weather stations from the 59 US counties with a large proportion of missing values over the period of time are discarded. Missing values in the retained weather stations are interpolated linearly between the available readings. Then, the weather data are summarized at the state level—the 59 counties are part of 19 different states. As a consequence, for each state, the 3 weather variables (*TMAX*, *TMIN* and *PREC*) are averaged over the retained weather stations of that state. Whatever the county considered, weather variables from all 19 states are utilized as covariates in ${\left\{\phi_{i}\right\}}_{i=1}^{N}$ of Equation (Equation 7). The final step before estimation is to transform all variables so that they are scaled and translated to achieve a unit marginal variance and zero mean.

Secondly, for the temporal patterns in the data, we consider three types: a week identifier (*WD*), a weight associated to each day of a week (*WD*) and a trend variable (*TREND*). The variable *WI* simply corresponds to a one-hot encoding of the week number of the year. The variable *WD* is added after observing that a regular pattern can be observed concerning the evolution of the CO${}_{2}$ emission with the day of the week—as shown in Figure 2, less emissions typically are observed during the weekend. The trend variable (*TREND*) is simply a linear and regularly increasing function at the daily rate from 0 (1 January 2010) to 1 (31 December 2015).

Finally, to take into account the time correlation of the CO${}_{2}$ emissions, we decided to use some lagged response variables as covariates. More precisely, after analyzing the autocorrelation function (ACF) of the time series of CO${}_{2}$ for each county (see Figure 3 for the ACF of three different counties), we proposed to use as covariates three lagged versions of the response variable. More precisely, for the *i*-th county at time *t*, $y_{t,i}$, the following lagged variables are used as predictors: the 365-day lagged variable $y_{t-365,i}$ (one year), the 182-day lagged variable $y_{t-182,i}$ (about six months) and the 14-day lagged variable $y_{t-14,i}$ (about 2 weeks).

### 4.2. Graph Construction of the Spatial Component

In this work, the 59 counties are considered the nodes of a common graph. The locations of the chosen counties are depicted in Figure 4. As a consequence, case 1 of the graph penalty function of Section 3.1 is considered, i.e., $\tilde{L}=I_{T}⊗L$. The single Laplacian matrix L is defined through the adjacency matrix.

A graph adjacency matrix should reflect the tendency for measurements made at node pairs to have similar values in mean. There are many possible choices for the design of this adjacency matrix. In this work, two different choices of matrix are compared. As in [11], we firstly construct the adjacency matrix based on distances by setting
(17)$$A_{i,j}^{dist}=e^{-l\frac{d_{i,j}^{2}}{\sum_{i,j}d_{i,j}^{2}}},$$
where $d_{i,j}$ denotes the geodesic distance between the *i*-th and *j*-th counties in kilometers and *l* is a scaling hyperparameter to be optimized using Algorithm 1. A heat map of the geodesic distances in kilometers between counties is represented in Figure 5.

The second proposition for the adjacency matrix is to utilize the empirical correlations between counties CO${}_{2}$ emissions. For two counties *i* and *j*, the adjacency coefficient is defined as follows:(18)$$A_{i,j}^{corr}=e^{-l\max\left(0,\;\rho_{i,j}^{2}\right)}$$
where $\rho_{i,j}$ is the empirical correlation between $y_{i}$ and $y_{j}$, the CO${}_{2}$ emissions of the *i*-th and *j*-th counties, respectively.

### 4.3. Numerical Experiments

In the following numerical experiments, the proposed penalized regression model over graph is compared to two other classical models, namely, the ridge and the ordinary least square (OLS) solution. In fact, these two models are nothing but special cases of the proposed model by setting in Equation (Equation 16) either $\gamma_{2}=0$ or $\left(\gamma_{1}=0,\gamma_{2}=0\right)$, respectively.

Firstly, we empirically study the performance of the penalized regression model over graph with the two possible choices for the Laplacian matrix. As shown in Table 1, using the adjacency matrix based on geodesic distances rather than on empirical correlations improves the RMSE on both the validation and the test sets. A smaller RMSE on the training set using the correlation-based adjacency matrix shows that this choice could lead to overfitting.

Table 2 shows the root mean squared error (RMSE) over the different sets (training, validation and test) with a varying number of training data. Let us remark as described more precisely in Section 3.2 that since we use the same number of data, increasing the size of training set reduces the size of both the validation and test sets. As expected, since the proposed model is a generalization of both the ridge and OLS solution, smaller RMSE is obtained on all configurations. More importantly, the proposed model allows us to obtain a quite significant improvement on the test set compared to both the ridge and the OLS solutions, which clearly demonstrates the superiority in terms of the generalization of the proposed model.

Next, in Table 3 we present the RMSE obtained when the models are applied without any lagged variables as covariates. By comparing the values obtained with these variables in Table 2, we can clearly see the benefit of using an auto-regressive structure in the regression model by the introduction of such lagged response variables.

In Figure 6, the weekly RMSE is depicted as a function of time for three different counties. These weekly RMSEs are obtained by aggregating the daily forecasted values from the proposed regression model which is trained on 50% of the dataset. It is interesting to observe that the weekly RMSE does not explode with time but rather stays quite stable with respect to time.

In order to ensure that the previously observed conclusions are not too sensitive to the specific 59 chosen counties, we compute the RMSE on the three different sets for the different regression models by randomly selecting 2 counties for each of the 19 states. Let remark that we use transfer learning for the hyperpamemeters of the models (i.e., $\gamma_{1}$ and $\gamma_{2}$). They are not optimized on each random choice of data but are set to their optimized values in the previous scenario in which all 59 counties are used. From the results depicted in Figure 7, the same conclusions as before can be drawn. It is worth noting that, even if the hyperparameters are not optimized for each random choice, the RMSE on the validation set is still smaller using the proposed model. Finally, the boxplots obtained on the test sets empirically show better predictive power for the proposed penalized regression model over graph prediction.

## 5. Conclusions

In this paper, we propose a novel GLM-based spatio-temporal mixed-effect model with graph penalties. This graph penalization allows us to take into account the inherent structural dependencies or relationships of the data. Another advantage of this model is its ability to model more complicated and realistic phenomena through the use of generalized linear models (GLMs). To illustrate the performance of our model, a publicly available dataset from the National Centers for Environmental Information (NCEI) in the United States of America is used, where we perform statistical inference of future CO${}_{2}$ emissions over 59 counties. We show that the proposed method outperforms widely used methods, such as the ordinary least squares (OLS) and ridge regression models. In the future, we will further study how to improve this model to this specific CO${}_{2}$ prediction. In particular, the use of different likelihood and link functions will be studied along with other adjacency matrices. We will also study whether considering, for the graph penalties, time differences instead of the direct mean values as discussed in Section 3.1 could improve the prediction accuracy. Finally, it will be interesting to connect this prediction model to some decision-making problems as in [22]. 

## Figures and Tables

**Figure 1 entropy-25-01272-f001:**
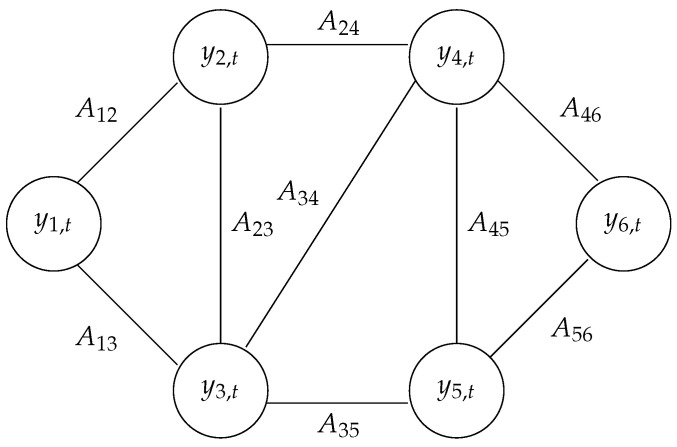
Example of a graph with |V|=6 vertices at time *t*.

**Figure 2 entropy-25-01272-f002:**
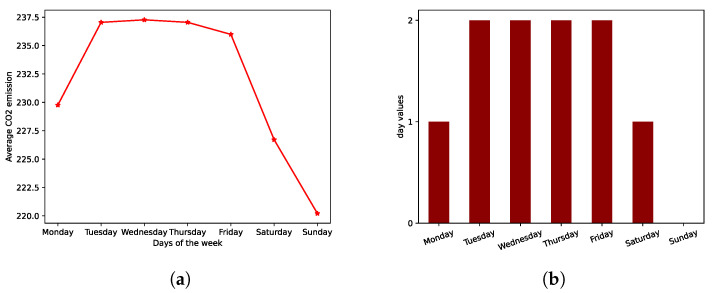
Choice of the covariate *WD* to encapsulate information about the weekday for the CO${}_{2}$ emission. (**a**) Spatial and temporal average of the CO${}_{2}$ emission per weekday. (**b**) Values assigned to the covariates *WD* depending on the current weekday.

**Figure 3 entropy-25-01272-f003:**
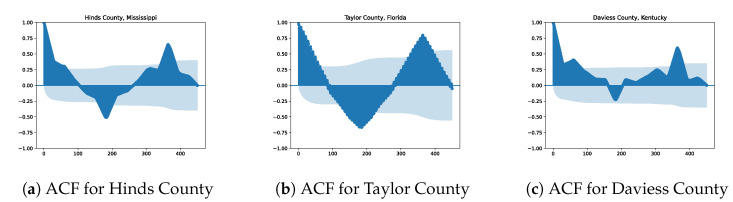
Illustration of the time correlation of the daily CO${}_{2}$ emissions per county with the autocorrelation function (ACF) of three different counties.

**Figure 4 entropy-25-01272-f004:**
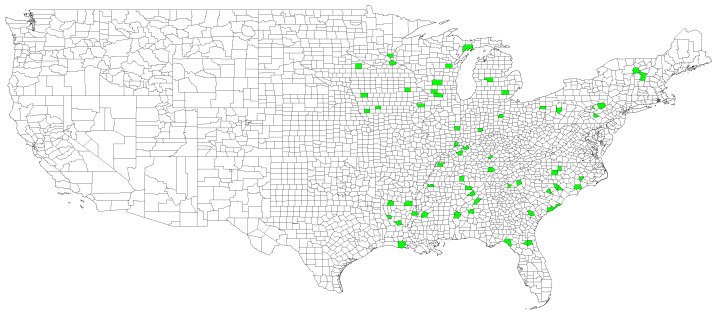
US counties selected as nodes of the graph depicted in green.

**Figure 5 entropy-25-01272-f005:**
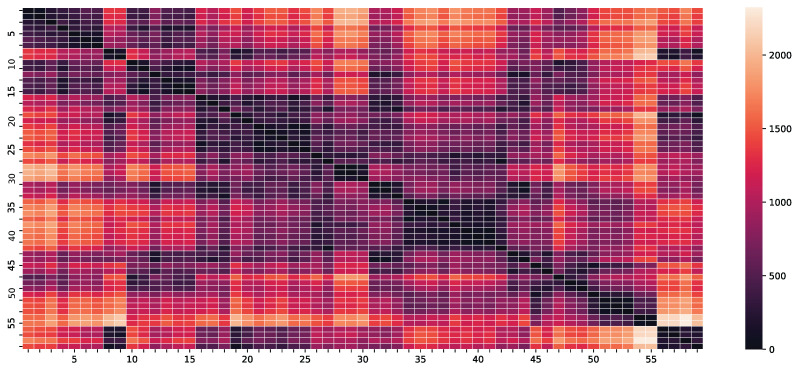
Geodesic distances in kilometers between counties.

**Figure 6 entropy-25-01272-f006:**
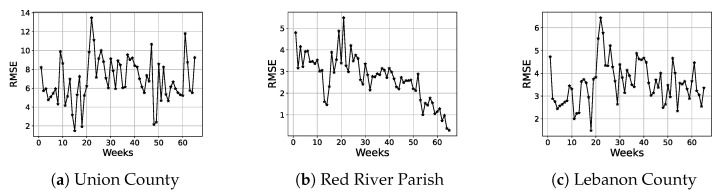
RMSE as a function of time for three different counties.

**Figure 7 entropy-25-01272-f007:**
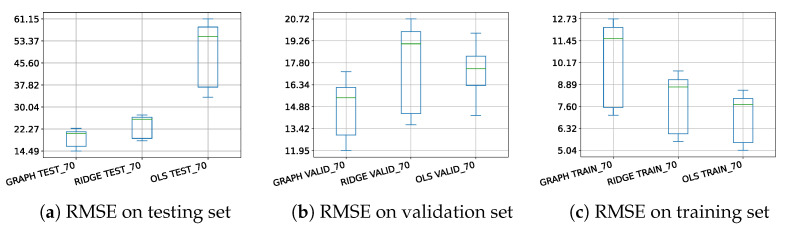
Boxplots of the RMSE obtained after 50 random choices of two counties per state for the different regression models (70% of the dataset is used for training).

**Table 1 entropy-25-01272-t001:** RMSE of the penalized regression model over graph with the Laplacian defined using an adjacency matrix based either on geodesic distances or on empirical correlations.

Root Mean Square Error (RMSE): Distances Versus Empirical Correlations						
	Testing Set		Validation Set		Training Set	
Perc. Train	Graph (Distance)	Graph (Correlation)	Graph (Distance)	Graph (Correlation)	Graph (Distance)	Graph (Correlation)
70%	**16.42**	27.04	**13.67**	14.92	13.40	**7.96**

**Table 2 entropy-25-01272-t002:** RMSE of the different regression models for different sizes of the training set.

Root Mean Square Error (RMSE)									
	Testing Set			Validation Set			Training Set		
Perc. Train	Graph Reg.	Ridge	OLS	Graph Reg.	Ridge	OLS	Graph Reg.	Ridge	OLS
50%	**35.65**	41.43	42.10	**16.80**	17.86	17.65	9.13	6.74	**6.55**
60%	**30.02**	36.77	41.41	**15.02**	19.60	19.73	21.73	**6.52**	**6.52**
70%	**16.42**	22.65	49.52	**13.67**	17.13	16.44	13.40	7.94	**7.02**

**Table 3 entropy-25-01272-t003:** RMSE of the different regression models without the use of the lagged response variables as covariates.

Root Mean Square Error (RMSE) without Lagged Variables									
	Testing Set			Validation Set			Training Set		
Perc. Train	Graph Reg.	Ridge	OLS	Graph Reg.	Ridge	OLS	Graph Reg.	Ridge	OLS
70%	**38.54**	**38.54**	41.76	**20.28**	**20.28**	20.34	9.65	9.65	**9.64**

## Data Availability

Data will be available on request.

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
