# Peer review of "Graph Regression Model for Spatial and Temporal Environmental Data—Case of Carbon Dioxide Emissions in the United States"

_entropy, 2023, doi:10.3390/e25091272_

Round 1
Reviewer 1 Report
The article introduces an interesting approach to forecast CO2-emissions. However, the paper still needs some clarifications and improvements.
1) lines 20-22. "due to human activities, the quantities of this gas have increased significantly, inducing global warming". Strictly speaking, the causal relationship still has not been proven. This is only one of the hypotheses, based on a correlation. One of the other causes of global warming could be, for example, earthquakes (see [https://doi.org/10.3390/geosciences12100372](https://doi.org/10.3390/geosciences12100372)).
2) line 26 I'd suggest replacing the word "fight" with "adaptation and mitigation measures"
3) A spell check is required. For example, line 199 "One the other hand" = "On the other hand"
4) Line 214. How did you select the 59 counties? Why them?
5) Font size in figures 6-7 could be bigger
6) The conclusion and discussion section should be extended. Try to describe for a wide range of readers why your model is better than other. You used data for 2010-2015. Were you able to predict emissions in 2016 and beyond? What was the prediction error? How does it differ from other models? Can your approach be applied to other countries? Are there other factors that should be included in the model or is it self-sufficient?
Author Response
Please see the attached pdf file.

Reviewer 2 Report
The manuscript entitled “Prediction of CO2 emissions in the United States via Graph Regression Model” (entropy-2521113) developed a model for predicting CO2 emissions in the United States by introducing Graph Regression and Generalized Liner Models. The content is meaningful and has high application value. However, there are still several points should be revised or addressed before the manuscript being accepted.
1- The title of this manuscript is “Prediction of CO2 emissions in the United States via Graph Regression Model”. From the title, this manuscript should be more focused on the predicted CO2 emission results in the US. However, this manuscript only briefly touches on the predicted CO2 emission results in Page 11-12, which mainly reflect the comparison among different methods. This may create a mismatch between the title fo the manuscript and its actual content. It is recommended to either revise the title to reflect the methogological focus or reallocate the content to incorporate a more thorough analysis of the predicted CO2 emission results, e.g., dynamic variation characteristics and spatial pattern of the predicted CO2 emissions in the US.
2- For the Introduction part, the second paragraph describes the research on the driving factors’ analysis of CO2 emissions in several countries. However, it appears that this paragraph lacks a strong connection to the prediction of CO2 emissions from both methodological and application perspectives. I would like to suggest removing this paragraph and including a literature review summarizing existing methodologies for CO2 prediction, as well as a summary of case studies predicting or analyzing the CO2 emissions in the US.
3- For the “1.1 Limitations of current methods and our contribution”, the proof to support the current methods’ limitation should be further strengthened.
4- This manuscript lacks the discussions on the deficiency of the new method and the research directions in the next step.
Author Response
Please see the attached pdf file

Round 2
Reviewer 1 Report
I have no further comments